# Public Willingness to Pay for Green Lifestyle in China: A Contingent Valuation Method Based on Integrated Model

**DOI:** 10.3390/ijerph20032185

**Published:** 2023-01-25

**Authors:** Jichao Geng, Na Yang, Wei Zhang, Li Yang

**Affiliations:** 1School of Economics and Management, Anhui University of Science and Technology, Huainan 232001, China; 2School of Economics and Management, Wenzhou University of Technology, Wenzhou 325000, China

**Keywords:** DBDC, WTP, integrated model, green lifestyle, carbon neutrality

## Abstract

The purpose of this study was to note how people recognize the green lifestyle and how much extra money they are willing to pay for it. An integrated model based on the theory of planned behavior was constructed, and data on the willingness to pay (WTP) for green lifestyles of 1377 respondents in five cities in East China were measured and calculated using the double-bounded dichotomous choice contingent valuation method (DBDC) combined with interval regression analysis. The results showed that the respondents were willing to pay an extra 81.8 yuan, 52.5 yuan, 38.9 yuan, 53.2 yuan, and 37.2 yuan per month for green food, clothing, travel, housing, and waste recycling, respectively. Attitude and moral norms were the strongest predictors of WTP for green clothing, travel, housing, and waste recycling. Perceived behavior control, environmental awareness, personal habits, subjective knowledge, gender, income, and education can affect a few kinds of green lifestyle’s WTP.

## 1. Introduction

Carbon emissions from human production and life lead to man-made climate change. Environmental problems only arise when every aspect of human life changes, including the way we eat, shop, travel, and live. The decline in ecological and environmental quality requires a significant reduction in carbon dioxide emissions in all aspects of human life. Energy use and travel had a direct impact on individual carbon emissions, while housing operations, food, and clothing had an indirect impact on it [1]. In addition, it is estimated that household consumption accounts for about two-thirds of global GHG emissions [2]. In developed countries such as the UK, the proportion is 70–75% [3,4]; however, the proportion in China is 35–40% [4,5], which is set to increase as the living standards improve [6]. In 2004, China overtook the United States to become the world’s largest producer of municipal solid waste [7]. From the perspective of carbon emission reduction, individual behavior can greatly affect the process of carbon neutrality [8].

At the beginning of 2021, The State Council of China issued the “Guidelines on Accelerating the Establishment and Improvement of a Green, Low-carbon and Circular Development Economic System,” which clearly stated that by 2025, significant progress would be made in transforming the way we work and live in a green way, and by 2035, green ways of working and living will be widely adopted [9]. Green lifestyles mainly refer to how man and nature live in harmony. We can define it as some environmental behaviors such as carbon emission reduction and sustainable and green consumption in the fields of clothing, food, housing, and travel. By doing so, people will live their life in a natural, environmental-friendly, frugal and healthy way [10,11]. As statistics shows, 67% of consumers tried to make a positive impact on the environment through their daily activities in 2021 [12]. Therefore, taking action to reduce or avoid emissions from personal consumption is of great significance to ecological environment improvement, and the best way accomplish this is by changing our behaviors and the way we live [13]. Under the background of high population density, rapid economic development, high pressure on environmental protection, and the requirement for eco-friendly development, China proposed to achieve carbon neutrality by 2060 [14]. In this paper, how individual behaviors participated in carbon neutrality, the extent to which they participated, and how they affected the process of carbon neutrality was explored. Moreover, based on the Implementation Plan for Promoting Green Consumption developed by the National Development and Reform Commission of China and other relevant departments in 2021 [9], we conducted our survey on residents’ WTP for carbon neutrality and the influencing factors from five aspects such as green food, clothing, travel, housing, and waste recycling.

Previous studies on the influencing factors of WTP are not very comprehensive, and most of them are based on the theory of planned behavior or only consider the impact of demographic economics variables on WTP [15,16,17,18,19,20]. In addition, the current literature focuses on a certain field or product of green lifestyle and does not comprehensively measure and compare the differences in people’s WTP in various fields of green lifestyle [21,22,23]. Although the CVM is the most frequently adopted one among all environmental assessment methods, it still has flaws and biases, such as hypothesis bias, initial bias, information bias, and so on. Given that the situation is hypothesized, it is a bit hard for people to attach themselves to the actual situation, which sometimes may lead to potential overestimation or underestimation. In addition, existing studies on the WTP for carbon neutrality mainly focus on aviation and tourism, and there are few studies in the literature that comprehensively examine the potential carbon-neutral WTP in individual lifestyles. We argue that there is limited information on the market for green products in China and the psychological drivers of consumers and that more research is needed to enrich the literature. In particular, the relatively low consumption of green products, despite sustainability and healthy living becoming an important quality-of-life consideration in recent years, suggests the need to understand better the drivers of consumer green behavior in the Chinese context. Therefore, this paper proposed to address two research questions: the first question was what were the factors associated with Chinese consumers’ willingness to implement a green lifestyle? The second question was how much Chinese residents are really willing to pay for a green lifestyle.

Therefore, this study examined different green lifestyles in an integrated manner rather than in a single domain, comparing the differences in their influencing factors and actual willingness to pay. Based on the theory of planned behavior, it combined psychological factors from other theoretical models that may influence the implementation of residents’ green lifestyles and incorporated demographic variables to explore whether different green lifestyles were influenced by the same factors and which factors had a greater impact on green lifestyles. To address the bias of the CVM survey method and the overestimation or underestimation of willingness to pay, this study rationalized the questionnaire design by investigating the attitudes of 1377 respondents toward green lifestyles through DBDC and calculating the true value of willingness to pay from the unique perspective of personal involvement in carbon neutrality.

Through the results of the study, this paper made four new contributions to the knowledge of this research area. First, based on the theory of planned behavior and other psychological variables that may influence the WTP model, this paper established an integrated WTP model to enhance its explanatory power, which is a novel framework in green lifestyle research. Second, in this paper, the WTP of Chinese residents for carbon-neutral products was examined, and the differences in residents’ WTP for green lifestyles in different fields were compared horizontally from various aspects of daily life (food, clothing, travel, housing, and waste recycling). Targeted research could expand the scope of the research and make the findings more supportive of relevant policies. Third, the study focused on the green product market in six central cities in East China, a geographical context that had received limited academic attention in the past. Last, DBDC was used to measure residents’ WTP of five green lifestyles to make the research data more accurate and scientific.

The results of this study would provide a scientific basis for the correct assessment and understanding of the impact of individual behaviors on carbon neutrality and the formulation of relevant policies for the development of carbon-neutral products. In addition, the difference in willingness to pay for different green lifestyles helped the government/relevant companies to adjust their strategies to induce residents to change their existing lifestyles in order to increase their familiarity and inclination towards green lifestyles, thus expanding the market share of green product consumption in China.

The organizational structure of this paper is as follows: Section two introduces the theoretical framework and reviews the relevant literature. Section three describes the survey instrument used, the measurement method, and the econometric model used. The descriptive statistics of related variables and the results of the interval-data regression model are presented in Section four. The discussion and conclusions of the results are provided in Section five.

## 2. Theoretical Framework

### 2.1. Carbon-Neutral WTP

Carbon neutrality means having a balance between emitting carbon and absorbing carbon from the atmosphere in carbon sinks [24]. Individual carbon neutrality refers to the total amount of carbon dioxide emissions and greenhouse gas emissions directly or indirectly produced by individuals within a certain period of time in their work and study and life being offset through afforestation, energy conservation, and emission reduction, that is, the relatively zero emissions [25]. Carbon-neutral products contribute to reducing carbon dioxide emissions, but they are more expensive and require extra money from residents for carbon offsets [26]. Carbon-neutral WTP refers to the degree to which consumers are willing to pay for carbon-neutral products. Estimates of the green lifestyle WTP vary in different regions and countries. This depends on many factors, such as the economic development level, environmental awareness, cultural background, social customs, and so forth. There are many foreign studies on residents’ WTP for emission reduction [27,28]. For example, Mostafa et al. [29] investigated Egyptian consumers’ WTP for products with carbon labels, and the results indicated that income, age, gender, and education level had significant impacts on respondents’ WTP. Lopez et al. [30] collected and analyzed data on the WTP for air pollution reduction of 900 residents from 14 rural towns in Spain, and it demonstrated that those citizens who are younger, better educated, and with higher incomes were willing to pay more. Schwirplies et al. [31] found that people with higher incomes and at a younger age have notably higher WTP for carbon offsets of green travel modes. Currently, some scholars have investigated the WTP for emission reduction of Chinese residents. Huang et al. [32], combining WTP with the life cycle index, evaluated the scheme of transforming food waste into energy in China and discovered that income, age, gender, and education level all had significant impacts on WTP. Zhang et al. [33] investigated 409 residents in a community in Guangzhou on their WTP for green buildings and found that young respondents with comparatively lower incomes would prefer to pay more.

It can be seen that a majority of demographic characteristics variables influence WTP, such as gender, age, educational background, and income level, among others. However, the impact of demographic characteristics on WTP varies greatly in different literature studies. Therefore, on account of the above-mentioned literature, we strive to reveal the determining factors of WTP by taking WTP as a measurement scale of people’s willingness for carbon dioxide emission reduction and adding the demographic characteristics as one of the research variables. This will help formulate reasonable and feasible policies to deal with the degradation of ecological and environmental quality caused by high carbon emissions.

### 2.2. Green Lifestyle

Lazer [34] defined lifestyle as a systematic concept of life. In general, a lifestyle is a distinctive way of life adopted by an entire society or a specific small group of people. Consumer lifestyle applications are a frequent topic of discussion among researchers, including consumer and market-oriented approaches. Plummer [35] stated that lifestyle is used to segment the entire consumer population into different groups based on various characteristics. Todd et al. [36] also suggested that lifestyle reflects the different lifestyles and consumption patterns of consumers. Nie and Zepeda [37] argued that lifestyle helps in segmenting consumers and understanding their attitudes and motivations. Reynolds et al. [38] suggested the use of consumer lifestyle studies to assess specific consumer behaviors, especially consumer activities.

Green lifestyles encompass all aspects of individuals’ willingness to protect the environment in their daily lives. Axsen et al. [39] defined green lifestyles as individuals’ involvement in various types of pro-environmental tasks and activities and suggested that consumers’ green purchasing decisions predict consumers’ green lifestyles. Lai et al. [40] hypothesized that pro-environmental purchasing behavior is an individual’s environmentally concerned behavior part of an individual’s environmental concern behavior. A green lifestyle also refers to nature-responsible pro-environmental behaviors, for example, using environmentally and ecologically friendly green products [41].

### 2.3. Theory and Factors

The theory of planned behavior (TPB) model, first proposed by Ajzen [42], suggests that individual attitudes, subjective norms, and perceived behavior control over specific behaviors can help us better understand environmental behaviors. Attitude is a measurable indicator of whether a person is satisfied with the outcome of a particular behavior, while subjective norms are the degree to which the peer’s perceptions and recognition of a particular behavior are influenced. Perceived behavior control is the degree of personal control over the execution of certain actions [43]. The raw TPB model has been widely used in environmental studies to analyze various behavioral intentions and behaviors, such as participation in environmental activities [19,31], consumption of environmental goods [44], travel of air passengers [45], forest protection [46], wildlife conservation [31,47], and greenhouse gas emissions [48]. As time changes and different disciplines blend, the raw TPB model is no longer able to interpret and accurately judge individual behavior fully. The theory, upgraded by Ajzen, incorporated moral norms that represented the moral satisfaction obtained when performing a particular behavior. Tao et al. [49], employing the extended TPB and adding the individual moral norms variable, explored the influencing mechanism of carbon offset behavior of Chinese consumers. Based on the extended TPB, Kang et al. carried out a study on environmental-friendly and sustainable textiles and apparel. It turned out that the purchase intention would be affected by consumers’ product knowledge.

Schwartz’s norm activation model (NAM) is mainly used to predict and understand prosocial and altruistic behaviors. The core variables of the NAM consist of responsibility ascription, personal norms, and other variables. Responsibility ascription reflects people’s views on the relationship between man and nature, and it mainly aims to explore the extent to which it can be transformed into people’s specific environmental behaviors. Personal norms refer to the strong obligation to perform prosocial behaviors due to the internalized sense of responsibility, which can also be called “moral norms.” Lind [50] found that personal norms and situational factors were key indicators of the low-carbon transportation choice behavior of urban residents. Jakovcevic and Steg [51] believed that personal norms and responsibility ascription had an evident impact on the willingness of Argentine residents to travel with low-carbon transportation.

In Triandis’s theoretical model of interpersonal behavior, people’s actual behavior is jointly determined by willingness, habit, convenience, and other factors. In the study of pro-environmental behavior, the theory of interpersonal behavior is used to determine whether morals and habits influence consumers’ low-carbon consumption behavior. Therefore, the personal habit factor is added as the psychological factor affecting the WTP.

In contrast to previous studies, environmental awareness, moral norms, subjective knowledge, and personal habits have not been discussed together, although some researchers have studied WTP in conjunction with other theoretical models. Therefore, this study proposes to introduce an integrated model (as shown in Figure 1) to study the WTP of Chinese residents for adopting a green lifestyle. Thus, a research model of consumers’ willingness to pay for carbon-neutral products is constructed to evaluate the relationship between attitude, perceived behavior control, subjective norms, moral norms, personal habits, subjective knowledge, environmental awareness, and WTP. The assumptions are as follows:

**Hypothesis** **1** **(H1).**
*Subjective norms have a positive impact on WTP;*


**Hypothesis** **2** **(H2).**
*Perceived behavior control has a positive impact on WTP;*


**Hypothesis** **3** **(H3).**
*Attitude has a positive impact on WTP;*


**Hypothesis** **4** **(H4).**
*Environmental awareness has a positive impact on WTP;*


**Hypothesis** **5** **(H5).**
*Moral norms have a positive impact on WTP;*


**Hypothesis** **6** **(H6).**
*Personal habits have a positive impact on WTP;*


**Hypothesis** **7** **(H7).**
*Subjective knowledge has a positive impact on WTP.*


## 3. Material and Methods

WTP is the degree to which consumers are willing to pay for a specific product or service, that is, the price that individuals are willing to pay to enhance the quality of life. The level of WTP reflects the degree of preference for non-market goods and the degree of recognition of non-market goods’ value. Some scholars believe that WTP is the only reasonable way to express the value of all commodities and utilities, and the contingent valuation method (CVM) is to learn people’s WTP by constructing an imaginary market. Therefore, a hypothetical market should be constructed so that a conceivable environment is created for residents. In this way, residents can be provided with sufficient information and avoid the interference from different environmental policies on their judgment to a certain degree, and thus avoid the deviation of WTP. Hence, WTP can be measured and evaluated more accurately.

CVM, in the case of a hypothetical market, uses the principle of utility maximization through questionnaire investigation to reveal consumers’ preference for public goods or services whose market price is hard to determine and thus deduce the value quantity of products and services in the end. So far, CVM has become the most widely used method to evaluate the value of environmental public goods, given its flexibility and wide applicability. In previous studies, the commonly-used methods include open-ended questions (OE) [52], payment card (PC) [53,54], and dichotomous choice (DC) method [55]. Among these methods, the application of DC can deliver a relatively more accurate survey result, and WTP is closer to the real willingness of residents to pay [55].

According to the number of questions, DC can be subdivided into single-bounded and double-bounded. Compared with the SBDC, DBDC has been much preferred in recent years, as it can provide more information about the “real” WTP of respondents, minimize the bias of WTP, and yield more accurate WTP [56]. Another advantage of the DBDC format is its consistency with the consumer utility maximization framework. In fact, an individual would be willing to pay the bill after being aware that the amount he paid can be compensated by some utilities obtained by buying the product [15]. Hanemann et al. [57] proved that DBDC is more effective than SBDC. A great number of empirical studies have applied the DBDC to assess the public’s WTP for environmental public goods. Over the years, the application of DBDC in assessing environmental damage has gradually shifted towards assessing environmental protection, and it has been widely used to assess biodiversity, electricity and energy, climate change, green consumption, and waste management [55,58,59,60,61,62]. From Hanemann’s [63] hypothesis, individuals know their utility function with certainty. In addition, this model can fully simulate the consumption behavior in marketing activities during the research process and provide respondents with an amount of money at their disposal, which they can use to pay for environmental protection. In such a situation, they just need to check whether they are willing to pay the bid value when doing the questionnaires. Accordingly, respondents’ valuation bias for environmental goods will be evidently reduced. In addition, the initial bid may provide an anchor for those respondents who are still hesitating, which means there still exists a bias in their answer for the second question, and the final WTP may be overestimated. However, most deviations can be avoided if the questionnaire is designed correctly, such as explaining the product correctly and designing a reasonable bid value, thus encouraging respondents to show their largest WTP [64,65,66,67].

After defining the aim of the study, we introduced DBDC to design the questionnaire and collect data on residents’ WTP for five green lifestyles.

### 3.1. Data Collection

In this paper, a questionnaire survey was conducted among Chinese residents by means of online distribution. The major provincial capitals of East China were selected as the reference market, including five cities: Jinan, Nanjing, Hefei, Hangzhou, and Shanghai. The main reason was that East China is China’s economic core region, accounting for more than one-third of China’s GDP. The five cities are important to China’s political, economic, and social activities, and the diversity of lifestyle and the malleability of the future are more representative.

First of all, to make sure that the initial bidding data is scientific and avoid the starting point bias of the CVM survey [64], a payment card style pre-survey of 50 face-to-face questionnaires was conducted prior to the formal survey. There were 35 bid values in the pre-survey, which fell in seven ranges: 0–10, 11–20, 21–40, 41–60, 61–90, 91–120, and 121–150. Since the majority of respondents chose the interval between 0–20, followed by 21–60 and 61–150, the interval between 0–20 was set as 10, the interval between 21 and 60 was set as 20, and the interval between 61 and 150 was set as 30. Then the initial bid, the upper and lower bound, and the order of questions were adjusted in accordance with the pre-survey. On this basis, the questions in the questionnaire were modified and improved to reduce confusion and misunderstanding among respondents and information asymmetry so that the efficiency of this questionnaire will be enhanced. According to the results of the pre-survey, we finally set the initial bidding data into seven levels: “5 yuan”, “15 yuan”, “30 yuan”, “50 yuan”, “75 yuan”, “100 yuan” and “120 yuan”, and thus measure the willingness of respondents to pay. Based on respondents’ answers to the first question, DBDC required a higher bid value in the case of a “yes” answer and a lower bid value in the case of a “no” answer. Here, it was assumed that consumers’ WTP in the second period was not affected by the price offered in the first period [68,69]. The questionnaire was available online for three weeks (from 15 January to 5 February 2022). The questionnaire was designed and conducted online through the Questionnaire Star website. A link to the online survey was also sent to residents through social media channels (WeChat). A snowball sampling method was used to invite respondents to distribute questionnaires to their friends, relatives, and colleagues, which would economically ensure a large number of questionnaires. The sample was spread all over China. Finally, a total of 1782 questionnaires were collected. As the questionnaire was designed for effective WTP, 1377 valid questionnaires were finally screened out after question screening and manual inspection.

The basic characteristics of the sample are shown in Table 1. Among the 1377 questionnaires, males accounted for 41.4%, and females accounted for 58.6%. The cities in the sample covered the major provincial capitals in East China, with Nanjing accounting for 22.2%, Shanghai at 29.5%, Hefei at 23.5%, Hangzhou at 14.4%, and Jinan at 10.7%. The age composition was mainly young and middle-aged people, with 55.3% under 30 years old, 19.8% between 31 and 40 years old, and 24.9% over 40 years old. The overall education level was comparatively high, with 50.3% accounting for a college degree, 16.6% for a master’s degree or above, 27.2% for junior colleges, and 5.9% for high school or below. Monthly household income of 5000 to 20,000 yuan accounted for the majority, accounting for 64.7 percent. Among them, the proportion of less than 5000 yuan was 16.4%, and the proportion of 20,000 yuan was 19%.

### 3.2. Measure

The data collection survey in this paper consisted of three parts, including 34 questions (Table 2). The first part was the sociodemographic characteristics of the respondents, including gender, age, educational background, permanent city of residence, occupation, and monthly household income.

The second part was a five-level Likert scale measuring psychological variables such as attitude, perceived behavior control, subjective attitude, subjective norms, environmental awareness, personal habits, moral norms, and subjective knowledge. The answers were based on a five-point scale ranging from “strongly disagree” to “strongly agree.” The third part was, based on DBDC from five aspects of green food, clothing, travel, housing, and waste recycling, to calculate the monthly plan for a carbon-neutral personal budget. Respondents were first presented with the initial bid level for each green lifestyle. The process of the investigator selecting the bid is shown in Figure 2. If the answer was yes, asked about the higher value of the bid; if the answer was no, the lower value of the bid was asked.

In the final stage of the questionnaire, an open-ended question that yielded the maximum WTP amount was presented to respondents who said “no” to both the amount they initially proposed and the amount after reduction at the second time, asking if they would be willing to provide the maximum amount of money. When they said they would not pay at all (zero bid), they would be asked to explain why they would not pay. The specific reasons why respondents are not willing to pay are shown in Figure 3.

This study aims to explore the influencing factors of residents’ willingness to adopt a green lifestyle. In line with the theoretical basis and reasonable arguments, we designed several questions for each factor. Hence, the reliability and validity of the questionnaire were reinforced. Among them, the measurement items of psychological variables were adapted from Tao et al. [49]’s study on Chinese consumers’ willingness to volunteer carbon offsets, Elena et al. [79]’s study on the intention of environmentally friendly food and He et al. [80]’s study on WTP for municipal waste. In addition, a certain number of sample sizes were selected to test the reliability of seven main concepts, such as subjective norms, perceived behavior control, attitude, environmental awareness, personal habits, moral norms, and subjective knowledge.

### 3.3. Statistical Model for Estimating the WTP Function

To understand how these preferences varied with diverse consumer characteristics, regression analyses were conducted to testify to the different impact various demographic and psychological factors had on WTP. Different preferences that might arise from two bids were avoided in the experimental results. In this regard, analysis models such as Logit or Probit models were not recommended as they would sort the WTP in accordance with the ordinal model and ignore the upper and lower value of the interval, so using the interval regression model to analyze the correlation between socioeconomic characteristics and psychological factors and WTP would be much advisable. In contrast, since DBDC provides each participant with an estimated WTP in four possible ranges, it was suitable to apply interval regression. Interval regression is a truncated regression model, which is a more well-accepted version of the Tobit model [69,81,82], used when there is an interval on the dependent variable rather than a unique value for each observation. In addition, it also estimated the probability that the latent variable WTP exceeds the lower bound of the interval but is lower than the upper bound. In addition, interval regression has been widely used to estimate the net output value of different products, services, or natural resources [83,84,85,86]. The individual WTP results were gained via the random utility model, where if the utility obtained from buying this product is higher than that obtained from rejecting bids and not buying the product, respondents will maximize their utility by choosing to purchase the product at the relevant bid amount.

Assuming that individuals express their preference for the green lifestyle, we can conclude from their responses to the DBDC questions that their true WTP. $y_{i}$ lies within one of four possible ranges $L_{i}\leq y_{i}\leq U_{i}$, where $L_{i}$ and $U_{i}$ represent the lower and upper limits, respectively. The response probability of the four intervals is as follows: (a) interval data [$t_{0}$, $t_{1}$], [$t_{2}$,${\;t}_{0}$]; (b) left-censored data [0, $t_{2}$]; (c) right censored data [$t_{1}$, +∞]. Where: $y_{i}$ = true WTP, $t_{0}$ = initial bid. $t_{1,2}$ = second bid. In accordance with Hanemann (1984), the upper bid for “yes, yes” responses were not truncated because a ‘yes’ answer to the second bid was not indicative of a maximum WTP but rather a lower bound of WTP for that value. This study adhered to Hanemann’s recommendation that the integration should not be extended to include negative WTP values since WTP studies provided very poor approximations of negative WTP compensation. Instead, as Hanemann suggested, these participants were allocated a WTP of zero on the basis of a genuine indifference towards a green lifestyle.

Assume that the model is:$$y_{i}=x_{i\beta }+\varepsilon_{i}$$

$y_{i}$ is the continuous, unobserved, and underlying latent variable of WTP, $x_{i}$ represents the vector of predictor variables associated with respondents, β is the vector of coefficients on WTP to be evaluated, $\varepsilon_{i}$ ($\varepsilon \sim N\left(0,\sigma^{2}\right)$ is denoted as a random error component of unobserved factors.

The probability of the respondent’s WTP for a green lifestyle being between a given lower and upper bound is:$$\mathrm{Pr}\left\{{\mathrm{WTP}}_{i}\subseteq \left(B_{i}{}^{L},B_{i}{}^{U}\right)\right\}=\phi \left(Z_{i}{}^{L}\right)-\phi \left(Z_{i}{}^{U}\right)$$

$Z_{i}$ represents the standard normal random variable, ϕ is the standard normal cumulative distribution function with $Z_{i}{}^{L}=\frac{B_{i}{}^{L}-c+x_{i}{}^{\prime }\beta +\varepsilon_{i}}{\sigma }$ and $Z_{i}{}^{L}=\frac{B_{i}{}^{U}-c+x_{i}{}^{\prime }\beta +\varepsilon_{i}}{\sigma }$, and $B_{i}{}^{L}$, $B_{i}{}^{U}$ being the lower and upper bound of the respondents’ WTP for a green lifestyle.

## 4. Results

### 4.1. Descriptive Statistics

Seven variables, including attitude, perceived behavior control, subjective norms, environmental awareness, subjective knowledge, moral norms, and personal habits, were set in the questionnaire to study the respondents’ attitudes and behavior toward carbon neutrality, as shown in Figure 4. Among them, the variables of attitude, perceived behavior control, and subjective norms were calculated from the values of two items in the questionnaire. The seven psychological variables were classified according to “strongly disagree, disagree, neither agree nor disagree, agree and strongly agree.” Seeing that the variables of attitude, perceived behavior control, and subjective norms were obtained by averaging the answers of the two sub-questions, it could be concluded that the mean value of subjective norm, perceived behavior control, and attitude was 3.75, 3.84, and 3.69, respectively. This indicated that the respondents’ attitudes towards feeling much social pressure, the ease of implementing a green lifestyle, and taking part in environmental action were all between “neither agree nor disagree” and “agree.”

The first and second bid values and winning rates of WTP for a green lifestyle were analyzed, as shown in Table 3. It can be seen that the proportion of valid questionnaires supporting the five initial bids was mostly distributed at 10 yuan, 20 yuan, and 40 yuan. Among the different green lifestyles, respondents seemed to be more willing to pay a premium for green food, followed by green clothing, green housing, waste recycling, and finally, green travel. The probability of respondents answering “yes” to the initial bid value was 74%, 55%, 53%, 53%, and 47%, respectively. Unlike what we expected, the frequency of “yes” in the first bid did not decrease with the increase in the proposed bid amount in the experiment: most respondents refused to accept the higher bids and accepted the lower one. The same was also true for the second bid. In the second bid, similar to the first, respondents seemed more willing to pay a premium for green food, followed by green clothing, green housing, and finally, waste recycling and green travel.

### 4.2. Validity and Reliability Analysis

The Cronbach coefficient was 0.943, higher than 0.9, which indicates that the final questionnaire has quite a good reliability. At the same time, the validity, usually evaluated by principal component analysis and KMO test, and Bartlett test, was also tested. By means of SPSS, we worked out that the all-variable KMO statistic was 0.960, above 0.7, and Bartlett’s sphericity test value was 0.00, below 0.1. It proved that we could analyze variables via factor analysis. Factor analysis of all variables showed that the results of component classification were consistent with the design structure of the questionnaire.

### 4.3. Interval Regression Analysis

All variables were applied to the regression model in three steps. First, only the psychological variables from the theory of planned behavior, such as subjective norm, perceived behavior control, and attitude, were employed in Model 1; then, the four independent variables from the integrated model, such as subjective knowledge, personal habits, moral norms, and environmental awareness were input into Model 2; and last, the control variables of age, gender, monthly household income, and education were added in Model 3.

The results of the interval regression analysis were demonstrated in Table 4; the significant variables in the regression analysis were summarized in Table 5. For green food, clothing, travel, housing, and waste recycling, 3, 3, 4, 4, and 6 of the eleven variables were statistically significant, at least at the 10% level, respectively.

For green food, in Model 1, attitude and subjective norm had a significant difference at the level of 1%, which indicated that the more positive the attitude of respondents towards buying green food and the stronger the subjective norm, the higher their WTP. In Model 2, all psychological variables except subjective norms and subjective knowledge were statistically significant. In Model 3, after adding demographic economics variables, it was found that education, gender, monthly household income, and age were not significantly associated with WTP. Environmental awareness was the strongest predictor in this study, with a regression coefficient greater than all other factors, followed by personal habits and perceived behavior control. This suggested that as people became more positive about their environmental awareness, personal habits, and perceived behavior control of green food, their WTP increased.

For green clothing, in Model 1, only the attitude variable had a strong significance at the level of 1%. In Model 2, only moral norms showed significance at a 1% level for the four variables added. Different from green food, the demographic economics variables in Model 3 showed strong significance for young people aged 18–25 years and elderly people aged 51–60 years. Different from previous studies, there was a negative correlation in family monthly income: the higher the family monthly income, the lower the WTP among the people whose monthly household income is more than 5000. This phenomenon has also been confirmed in previous studies [87].

For green travel, in Model 1, as with green food, attitude (1%) and subjective norm (5%) both showed strong significance, and in Model 2, perceived behavior control (1%) and environmental awareness (5%) the newly added variable showed strong correlation. In Model 3, the subjective norm that was significant in Model 1 and Model 2 lost its significance. At the same time, environmental awareness that was significant in Model 2 turned out to be insignificant. It was finally concluded that the younger the age, the stronger the WTP. This could be explained as that the older have a longer life expectancy than the younger ones, and they may therefore benefit more from good health. Another explanation was that most of the young people’s travel mode was based on public transportation, and they had no greater ability and willingness to change their travel habits. In contrast, there was no correlation shown between WTP and education, gender, or monthly household income.

For green housing, in Model 1, only the attitude variable had a strong significance at the level of 1%, while neither subjective norm nor perceived behavior control showed significance. In Model 2, the newly added moral norms and subjective knowledge were significant at the level of 1% and 5%, respectively. What is different in Model 3 is that women, except for other green lifestyles, were more willing to pay an extra 5.1 yuan per month for green housing than men. However, age, education, and monthly household income were not significantly correlated.

For waste recycling, in Model 1, attitude (1%) had a considerable positive impact. It suggested that the more positive the attitude towards waste recycling one held, the more willing one would be to pay more fees for it. Then was the perceived behavior control variable (5%). In Model 2, among these four additional variables, only environmental awareness (10%) and moral norms (1%) were statistically significant. This result meant that the more concerned the respondents were about the environment and the more environmentally responsible they were, the higher their WTP for waste recycling would be. In Model 3, respondents aged 18–25 might be more willing to perform waste recycling than those aged under 18. Interestingly, the higher the monthly household income, the lower the WTP for waste recycling among people whose monthly household income is more than 5000. From these results, it could be concluded that H3, H4, and H7 were confirmed, and H1, H2, H5, and H6 were rejected for the full DC WTP model.

We used the Stata module “doubleb” written by Lopez-Feldman to estimate the mean WTP, and the results are shown in Table 6. The mean WTP for green food, clothing, travel, housing, and waste recycling were 81.8 yuan, 52.5 yuan, 38.9 yuan, 53.2 yuan, and 37.2 yuan, respectively.

## 5. Discussion

This study proposed to investigate the willingness to pay for green lifestyles of consumers in six major central cities in East China using the DBDC survey method. Analysis of the willingness-to-pay data collected from 1377 respondents showed that most of the associations of the study’s hypotheses were supported, and the differences in the average willingness-to-pay for different green lifestyles were confirmed in previous studies.

Among the control variables, age, education level, monthly household income, and gender all had an impact on the value of WTP for a green lifestyle. A green lifestyle, especially green travel, was positively influenced by age. Compared with respondents aged under 18, the older the respondents, the more willing they were to pay for green travel. This could be explained as that the older have a longer life expectancy than the younger ones, and they may therefore benefit more from good health. Another explanation was that most of the young people’s travel mode was based on public transportation, and they had no greater ability and willingness to change their travel habits. Income played an important role in the WTP for green clothing and waste recycling in China. This study pointed out that when monthly household income was less than 5000 yuan, the income had little effect on the WTP. When the monthly household income was more than 5000 yuan, the higher the income was, the lower the WTP was. It indicated that people with a monthly income of less than 5000 yuan might be the ones who pay the largest premium for green clothing and waste recycling in China in 2021. They expected a green life of higher quality. Women were willing to pay an extra 5.2 yuan per month for green housing than men. Surprisingly, highly educated people were less willing to pay more per month for green travel than their less educated counterparts.

The perceived behavior control, attitude, personal habits, moral norms, environmental awareness, and subjective knowledge in the integrated model were significant predictors of WTP for a green lifestyle. Among them, attitude and moral norms were the strongest predictors of four green lifestyles (green clothing, travel, housing, and waste recycling), indicating the more willing and responsible residents were to protect the environment, the more willing they were to pay a premium for green products. Perceived behavior control had an impact on WTP for green food, travel, and waste recycling. Environmental awareness, personal habits, and subjective knowledge can only affect a few kinds of green lifestyles’ WTP (Environmental awareness can affect green food and waste recycling; personal habits can affect green food; subjective knowledge can affect green housing). Only subjective norms had no influence on all green lifestyles. The inclusion of these variables improved the predictive power of the theoretical framework in determining WTP. This could yield interesting results, as they gave accurate insight into WTP.

Based on the final calculated average willingness to pay, it can be seen that Chinese residents were more willing to pay more for a green lifestyle that can bring short-term personal benefits (green food, clothing, and housing), while they were less willing to pay for a green lifestyle that might benefit them in the long term (green travel and waste recycling). Food, clothing, and housing are more central areas to guide residents to adopt a green lifestyle.

Some psychological and demographic variables found in this study that could predict residents’ WTP for a green lifestyle were supported by some literature studies. For example, in terms of green food, the research of Cooper et al. [88] showed that almost two-thirds of Australian shoppers were willing to pay higher prices for health-starred products. Huang et al. [18] found that Taiwanese consumer respondents were willing to pay an additional $21.95 per year for organic CAS milk. As to green clothing, Jacobs et al. [89], based on a large number of samples of German female consumers, investigated the possible contributing factors and the influence of barriers to sustainable clothing purchasing behavior and spotted that a positive attitude of altruistic value could promote the purchase of sustainable clothing. As regards green housing, Zalejska-Jonsson [90] studied Swedish residents’ WTP for green buildings, and the results revealed that people were willing to pay more for low-energy buildings. There were many literature studies showing that [31,91] travel and waste recycling were indeed the difficulties and obstacles of deep carbon emission reduction for residents, and people were quite critical of carbon-neutral products in these two fields.

## 6. Conclusions

A green lifestyle is a sustainable and green environmental behavior; however, the implementation of green lifestyles is not yet widely spread. Therefore, investigating residents’ willingness to pay for green lifestyles could motivate all parties in society to create an environment for green consumption and thus motivate residents to implement green lifestyles, which could also contribute to achieving carbon neutrality at the individual level to some extent. Our study aimed to empirically investigate the differences in consumers’ intention to purchase green products and their true willingness to pay for different green lifestyles. With regard to question one, our results confirmed that attitude and moral norms were the strongest predictors of WTP for green clothing, travel, housing, and waste recycling. Perceived behavior control had an impact on WTP for green food, travel, and waste recycling. Environmental awareness, personal habits, and subjective knowledge could only affect a few kinds of green lifestyle WTP. For women, they were willing to pay more for green housing; for the older and the less educated ones, they paid more for green travel; for those higher-income individuals who earned more than 5000 yuan per month, they paid less for green clothing and waste recycling. Coming to question two, the results of our study suggested that the average WTP of respondents for green food, clothing, travel, housing, and waste recycling were 81.8 yuan, 52.5 yuan, 38.9 yuan, 53.2 yuan, and 31.2 yuan per month, respectively.

### 6.1. Theoretical Implications

This paper provided four main contributions to the theoretical knowledge in this research area. First, an integrated theoretical framework was proposed to enrich the drivers that influence residents’ willingness to pay for a green lifestyle. Implementing a green lifestyle was considered to be a sustainable and healthy consumption behavior, and the comprehensive theoretical model proposed in this paper could provide a broader scope for future research to investigate the intrinsic drivers of residents’ implementation of a green lifestyle, thereby promoting more sustainable green consumption patterns. Second, this paper provided a detailed classification of green lifestyles. However, the current literature has not yet examined the variability of green consumption behaviors from this research perspective, which could contribute to future research studies examining the variability of different green lifestyles. Third, the study highlighted the behavior of implementing green lifestyles in China, and the findings filled a gap in the literature regarding the differences in Chinese consumers’ willingness to implement different green lifestyles. Finally, the use of the DBDC method for studying different green lifestyles could extend the research area of DBDC survey methods, which could also contribute to the literature on survey methods for studying green lifestyles in the future.

### 6.2. Practical Implications

Although respondents were willing to pay a premium for a green lifestyle, it did not indicate that they would buy green products in real life. A better understanding of the reasons behind rejection could also help improve residents’ WTP and allow a better job of policy adjustment and publicity guidance. Therefore, policymakers should understand the motivation behind the WTP and carry out environmental protection propaganda and environmental policy implementation in a rational and targeted way. In order to convince consumers of the quality and safety of green food and clothing, as well as the benefits of green travel, housing, and waste recycling, the government should support and encourage organizations or institutions that provide certification and labeling for green food and clothing and popularize the knowledge of green housing, travel, and waste recycling. In addition, the research results on the influence of consumer factors on the WTP of green products also provided important information to enterprises. A clear label with complete information about green products would help them better understand the features and benefits of green products, as well as take full advantage of consumer demand and expand green market share. Understanding the residents’ WTP on green lifestyles helps companies design better marketing strategies to stimulate residents to buy green products, thus expanding the market share of green product consumption in China. In addition, it can also help the government to develop reasonable incentive policies to promote the implementation of green lifestyles among residents. It can also be used to achieve important social benefits, such as reducing carbon emissions.

### 6.3. Limitations and Future Scope

The results of the study could be used as a source of information for the academic field of social research, as well as for the government to build consumer trust in the long-term benefits of implementing a green lifestyle and to develop sustainable marketing strategies for relevant companies. However, there were still limitations in this study. First, this study might not be comprehensive enough in its delineation of green lifestyles, examining the variability of their willingness to pay in only five areas. Second, since we could only analyze our model on account of those who express their WTP greater than zero, our estimates would be affected by selection bias, social approval bias, and so on. Finally, considering the nature of the sampling process, it might be difficult to include some groups. Due to the small proportion of low-educated people in our survey, they might overestimate their WTP, and there might be limitations in the online questionnaire survey.

Therefore, future studies could be more refined and comparative in green lifestyles to provide a more comprehensive scientific examination of the topic. Studies showed that there were several types of biases in CVM: social approval bias, sample frame bias, affinity bias, and non-response error. Therefore, future studies may study the impact of several biases on WTP for a green lifestyle and may use a multi-boundary WTP format to obtain more effective results. Finally, future researchers should conduct more studies to investigate the reasons why consumers resist implementing a green lifestyle and focus on analyzing non-purchasers pre-adoption resistance to the consumption of green products.

## Figures and Tables

**Figure 1 ijerph-20-02185-f001:**
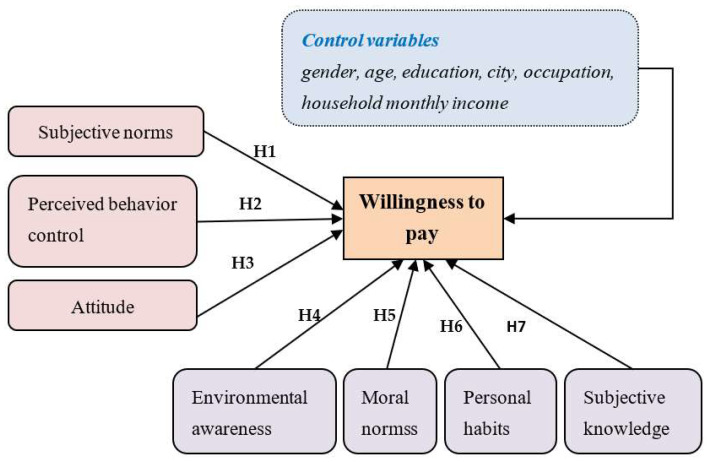
Theoretical model.

**Figure 2 ijerph-20-02185-f002:**
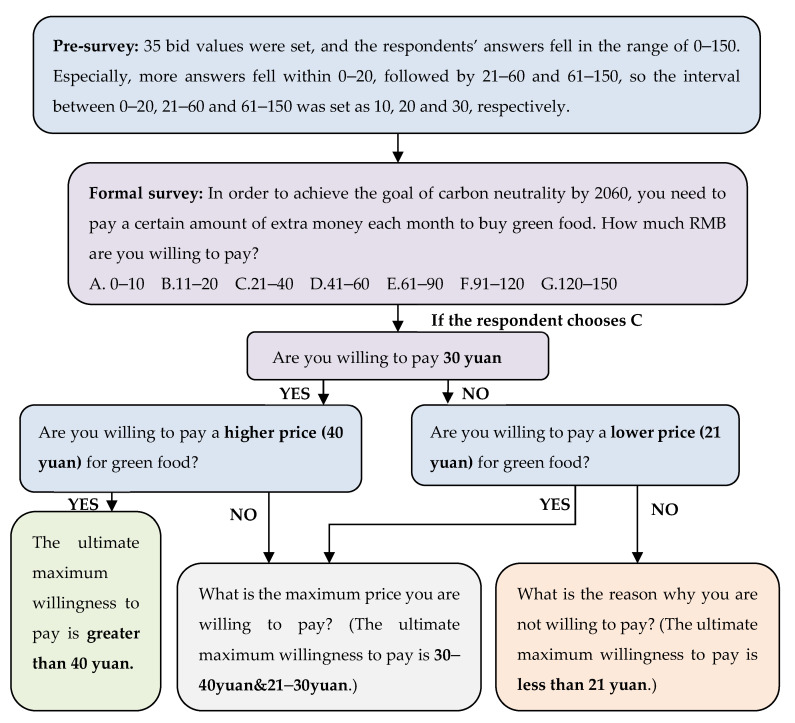
Bidding process (Take green food, for example).

**Figure 3 ijerph-20-02185-f003:**
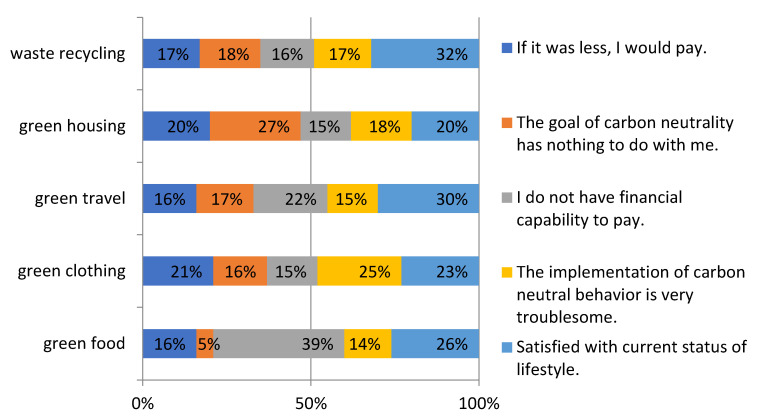
Reasons for rejecting the offered bids.

**Figure 4 ijerph-20-02185-f004:**
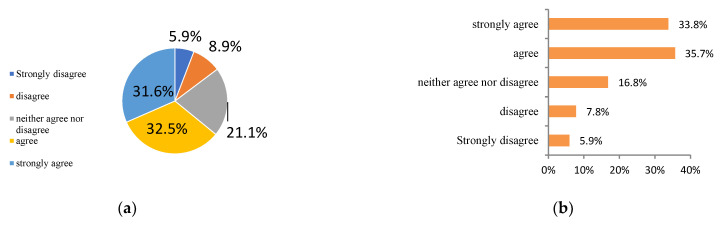
Statistical analysis of psychological variables. (**a**) Environmental awareness, (**b**) Moral norms, (**c**) Personal habits, (**d**) Subjective knowledge.

**Table 1 ijerph-20-02185-t001:** Sample descriptive statistics.

Items	Frequency (*n* = 1377)	Relative Weight (%)
Gender		
Male	570	41.4
Female	807	58.6
Age		
<18	24	1.7
18–25	414	30.1
26–30	324	23.5
31–40	273	19.8
41–50	315	22.9
51–60	27	2.0
Education		
Incomplete secondary school	39	2.8
Secondary school	42	3.1
Junior College	375	27.2
Bachelor degree	693	50.3
Master	228	16.6
Household income (RMB per month)		
<2000	63	4.6
2001–5000	162	11.8
5001–10,000	387	28.1
10,001–20,000	504	36.6
>20,000	261	19.0
City		
Nanjing	306	22.2
Shanghai	402	29.2
Hefei	324	23.5
Hangzhou	198	14.4
Jinan	147	10.7

**Table 2 ijerph-20-02185-t002:** Measurement items of variables.

Dependent Variables [Reference]	Item
DC WTP	In order to achieve the goal of carbon neutrality by 2060, you need to pay a certain amount of extra money each month for green food/clothing/travel/housing/waste recycling. Are you willing to pay “xx”? (RMB 5, 15, 30, 50, 75, 100, 120) and after a DC question (Yes/No), the initial bid was increased or decreased.
Independent variables	Item(s) in the questionnaire
Attitude	Indicate the level of agreement with the following statements:
[70,71]	Item 1: I think it is very wise to adopt a green lifestyle *.
	Item 2: I think the implementation of a green lifestyle can achieve energy conservation and emission reduction *.
Subjective norms	Indicate the level of agreement with the following statements:
[71,72]	Item 1: My friends and family expect me to adopt a green lifestyle *.
	Item 2: My family, friends, and people around me are adopting a green lifestyle *.
Perceived behavior control	Indicate the level of agreement with the following statements:
[71]	Item 1: I have many opportunities to practice a green lifestyle *.
	Item 2: It is entirely up to me to implement a green lifestyle *.
Environmental awareness	Indicate the level of agreement with the following statements:
[73,74,75]	I am always concerned about environmental protection in my daily life *.
Moral norms	Indicate the level of agreement with the following statements:
[76]	In my daily life, I have the responsibility to implement energy-saving and emission-reduction behaviors *.
Personal habits	Indicate the level of agreement with the following statements:
[20]	I have been used to energy conservation and emission reduction in my daily life *.
Subjective knowledge	Indicate the level of agreement with the following statements:
[77,78]	I know how to achieve energy conservation and emission reduction in daily life.*
Control variables	Item
Gender	Gender of respondent (1 = male; 2 = female)
Age	Age of respondent (1) Less than 18; (2) 18–25; (3) 26–30; (4) 31–40; (5) 41–50; (6) 51–60; (7) More than 60
Monthly household income	In what category does the total monthly income of your household fall? (1) Less than RMB 2000;(2) RMB 2001–5000; (3) RMB 5001–10,000; (4) RMB 10,001–20,000; (5)more than RMB 20,000
Education	What is your educational background? (1) Incomplete secondary school; (2) Secondary school; (3) Junior College; (4) Bachelor degree; (5) Master

* = 5-point Likert scale: 1 = strongly disagree; 3 = neither agree nor disagree; 5 = strongly agree.

**Table 3 ijerph-20-02185-t003:** Answer to bids.

Bid Cards *	Bid Card Statistics								Yes To First Bid								NO To First Bid				
	Green Food		Green Clothing	Green Travel	Green Housing		Waste Recycling		Green Food		Green Clothing	Green Travel	Green Housing		Waste Recycling		Green Food	Green Clothing	Green Travel	Green Housing	Waste Recycling
5/1/10	216 (16)		276 (20)	195 (14)	189 (14)		444 (32)		132 (61)		123 (45)	84 (43)	87 (46)		183 (41)		84 (39)	153 (55)	111 (57)	102 (54)	261 (59)
15/10/20	261 (19)		291 (21)	273 (20)	231 (17)		345 (25)		183 (70)		135 (46)	93 (34)	117 (51)		150 (43)		78 (30)	156 (54)	180 (66)	114 (49)	195 (57)
30/20/40	276 (20)		183 (13)	300 (22)	297 (22)		246 (18)		189 (68)		123 (67)	153 (51)	117 (39)		156 (63)		87 (32)	60 (33)	147 (49)	180 (61)	90 (37)
50/40/60	255 (19)		210 (15)	309 (22)	258 (19)		147 (11)		207 (81)		117 (56)	123 (40)	147 (57)		87 (59)		48 (19)	93 (44)	186 (60)	111 (43)	60 (41)
75/60/90	63 (5)		210 (15)	93 (7)	120 (9)		66 (5)		54 (86)		102 (49)	54 (58)	72 (60)		45 (68)		9 (14)	108 (51)	39 (42)	48 (40)	21 (32)
100/90/120	216 (16)		132 (10)	156 (11)	213 (15)		78 (6)		183 (85)		90 (68)	96 (62)	129 (61)		66 (85)		33 (15)	42 (32)	60 (38)	84 (39)	12 (15)
150/120/180	90 (7)		75 (5)	54 (4)	69 (5)		51 (4)		69 (77)		63 (84)	42 (78)	63 (91)		48 (94)		21 (23)	12 (16)	12 (22)	6 (9)	3 (6)
Total	1377		1377	1377	1377		1377		1017 (74)		753 (55)	645 (47)	732 (53)		735 (53)		360 (26)	624 (45)	732 (53)	642 (47)	1017 (74)
Answer to Bids																					
Green Food				Green Clothing				Green Travel				Green Housing				Waste Recycling					
YY		NY		YY		NY		YY		NY		YY		NY		YY		NY			
YN		NN		YN		NN		YN		NN		YN		NN		YN		NN			
96 (7.0)		45 (3.3)		69 (5.0)		78 (5.7)		54 (3.9)		36 (2.6)		60 (4.4)		30 (2.2)		99 (7.2)		102 (7.4)			
36 (2.6)		39 (2.8)		54 (3.9)		75 (5.4)		30 (2.2)		75 (5.4)		27 (2.0)		72 (5.2)		87 (6.3)		156 (11.3)			
96 (7.0)		45 (3.3)		93 (6.8)		75 (5.4)		54 (3.9)		66 (4.8)		57 (4.1)		33 (2.4)		75 (5.4)		84 (6.1)			
87 (6.3)		33 (2.4)		42 (3.1)		81 (5.9)		39 (2.8)		114 (8.3)		60 (4.4)		81 (5.9)		75 (5.4)		111 (8.1)			
114 (8.3)		45 (3.3)		90 (6.5)		24 (1.7)		102 (7.4)		36 (2.6)		90 (6.5)		54 (3.9)		111 (8.1)		27 (2.0)			
75 (5.4)		42 (3.1)		33 (2.4)		36 (2.6)		51 (3.7)		111 (8.1)		27 (2.0)		126 (9.2)		45 (3.3)		63 (4.6)			
138 (10.0)		27 (2.0)		75 (5.4)		30 (2.2)		84 (6.1)		54 (3.9)		102 (7.4)		39 (2.8)		63 (4,6)		24 (1.7)			
69 (5.0)		21 (1.5)		42 (3.1)		63 (4.6)		39 (2.8)		129 (9.4)		45 (3.3)		72 (5.2)		24 (1.7)		36 (2.6)			
48 (3.5)		0 (0.0)		66 (4.8)		36 (2.6)		36 (2.6)		12 (0.9)		42 (3.1)		18 (1.3)		45 (3.3)		3 (0.2)			
6 (0.4)		9 (0.7)		36 (2.6)		72 (5.2)		18 (1.3)		27 (2.0)		30 (2.2)		30 (2.2)		0 (0.0)		18 (1.3)			
144 (10.5)		21 (1.5)		72 (5.2)		3 (0.2)		75 (5.4)		9 (0.7)		90 (6.5)		36 (2.6)		60 (4.4)		6 (0.4)			
36 (2.6)		15 (1.1)		18 (1.3)		39 (2.8)		21 (1.5)		51 (3.7)		39 (2.8)		48 (3.5)		6 (0.4)		6 (0.4)			
57 (4.1)		12 (0.9)		48 (3.5)		6 (0.4)		36 (2.6)		6 (0.4)		51 (3.7)		0 (0.0)		36 (2.6)		3 (0.2)			
18 (1.3)		3 (0.2)		12 (0.9)		9 (0.7)		6 (0.4)		6 (0.4)		12 (0.9)		6 (0.4)		9 (0.7)		3 (0.2)			

* The first value represents the first bid, while the second and third values represent the upper and lower bids, respectively. YY = (yes, yes) response, YN = (yes, no) response, NY = (no, yes) response, NN = (no, no) response. The percentages are in parentheses; Values outside the parentheses are frequency.

**Table 4 ijerph-20-02185-t004:** Interval regression results to green lifestyles.

Variables	Dependent Variable: WTP Extra for Green Food				Dependent Variable: WTP Extra for Green Clothing				Dependent Variable: WTP Extra for Green Travel				
	Model 1		Model 2	Model 3	Model 1	Model 2		Model 3	Model 1		Model 2		Model 3
Constant	77.125 (0.000)		76.457 (0.000)	92.539 (0.000)	55.624 (0.000)	55.619 (0.000)		56.959 (0.000)	48.259 (0.000)		2.131 (0.363)		48.274 (0.000)
Subjective norms	8.404 (0.004)		0.292 (0.926)	0.129 (0.967)	−1.019 (0.661)	−2.702 (0.289)		−1.622 (0.521)	4.561 (0.032)		−4.53 (0.046)		1.794 (0.442)
Perceived behavior control	−1.100 (0.702)		−6.695 (0.026)	−6.933 (0.023)	−0.525 (0.820)	−1.258 (0.607)		−1.136 (0.643)	−2.734 (0.199)		11.954 (0.000)		−3.922 (0.086)
Attitude	11.564 (0.000)		6.373 (0.022)	3.584 (0.225)	17.835 (0.000)	16.666 (0.000)		10.874 (0.000)	13.679 (0.000)		3.356 (0.092)		8.683 (0.000)
Environmental awareness			10.530 (0.000)	10.943 (0.000)		1.707 (0.434)		2.556 (0.239)			4.893 (0.012)		2.358 (0.235)
Moral norms			4.313 (0.099)	3.994 (0.125)		6.693 (0.002)		5.173 (0.015)			−0.975 (0.606)		3.856 (0.047)
Personal habits			5.848 (0.019)	6.192 (0.014)		−1.616 (0.431)		−0.624 (0.759)			−0.326 (0.863)		−0.159 (0.933)
Subjective knowledge			2.288 (0.360)	2.774 (0.265)		−2.816 (0.176)		−1.417 (0.487)					0.254 (0.892)
Gender (female)				3.619 (0.285)				−2.436 (0.385)					2.023 (0.433)
Education				−3.014 (0.181)				−1.179 (0.501)					−6.829 (0.000)
Age (<18)				/				/					/
Age (18–25)				8.833 (0.561)				33.197 (0.001)					41.467 (0.000)
Age (26–30)				−4.178 (0.782)				13.777 (0.181)					25.048 (0.008)
Age (31–40)				−7.895 (0.602)				7.468 (0.469)					21.656 (0.021)
Age (41–50)				−9.805 (0.514)				10.026 (0.327)					15.061 (0.106)
Age (51–60)				22.070 (0.272)				26.358 (0.073)					23.251 (0.084)
Monthly household income (<2000)				/				/					/
Monthly household income (2001–5000)				−6.461 (0.500)				−1.466 (0.854)					11.824 (0.108)
Monthly household income (5001–10,000)				−7.056 (0.418)				−15.207 (0.036)					−2.204 (0.736)
Monthly household income (10,001–20,000)				−6.898 (0.432)				−13.092 (0.071)					−7.448 (0.256)
Monthly household income (>20,000)				−0.475 (0.958)				−18.238 (0.016)					2.822 (0.679)
Log-Likelihood	−2624.4863		−2605.5201	−2594.9236	−2875.8237	−2869.6227		−2829.465	−2877.0896		−2871.7617		−2824.4236
Number of observations	1377		/	□/	1377	□/		□/	□1377		□/		□/
Variables		Dependent Variable: WTP Extra For Green Housing					Dependent Variable: WTP Extra For Waste Recycling						
		Model 1		Model 2	Model 3		Model 1			Model 2		Model 3	
Constant		56.67 (0.000)		59.751 (0.000)	73.741 (0.000)		42.322 (0.000)			42.262 (0.000)		35.782 (0.004)	
Subjective norms		3.089 (0.210)		1.833 (0.499)	2.802 (0.296)		2.512 (0.248)			−0.125 (0.958)		0.916 (0.692)	
Perceived behavior control		1.509 (0.533)		2.052 (0.427)	0.944 (0.714)		−4.537 (0.035)			−5.976 (0.009)		−5.620 (0.012)	
Attitude		13.461 (0.000)		12.628 (0.000)	8.096 (0.001)		17.983 (0.000)			16.283 (0.000)		9.326 (0.000)	
Environmental awareness				−2.831 (0.213)	−2.854 (0.204)					3.525 (0.078)		4.057 (0.038)	
Moral norms				7.118 (0.001)	6.115 (0.006)					5.495 (0.007)		3.697 (0.058)	
Personal habits				2.492 (0.248)	3.004 (0.159)					−0.667 (0.727)		0.965 (0.604)	
Subjective knowledge				−5.184 (0.015)	−3.810 (0.069)					−1.634 (0.383)		−0.481 (0.790)	
Gender (female)					5.142 (0.079)							0.418 (0.869)	
Education					−2.296 (0.210)							−0.052 (0.974)	
Age (<18)		/		/	/		/			/		/	
Age (18–25)					11.718 (0.341)							37.632 (0.000)	
Age (26–30)					−8.935 (0.464)							14.436 (0.113)	
Age (31–40)					−19.347 (0.112)							9.930 (0.277)	
Age (41–50)					−15.935 (0.189)							4.205 (0.642)	
Age (51–60)					−18.408 (0.252)							19.048 (0.134)	
Monthly household income (<2000)		/		/	/		/			/		/	
Monthly household income (2001–5000)					11.757 (0.159)							0.021 (0.998)	
Monthly household income (5001–10,000)					−3.924 (0.594)							−11.393 (0.083)	
Monthly household income (10,001–20,000)					−6.365 (0.389)							−14.802 (0.024)	
Monthly household income (>20,000)					3.468 (0.653)							−13.457 (0.050)	
Log-Likelihood		−2850.6866		−2841.6813	−2794.482		−3349.181			−3342.927		−3271.7927	
Number of observations		1377		/	/		1377			/		/	

The *p* values are in parentheses; Values outside the parentheses are the coefficients.

**Table 5 ijerph-20-02185-t005:** Significant variables.

Variables	Green Food	Green Clothing	Green Travel	Green Housing	Waste Recycling
Subjective norms	-	-	-	-	-
Perceived behavioral control	significant, negative	-	significant, negative	-	significant, negative
Attitude	-	significant, positive	significant, positive	significant, positive	significant, positive
Environmental awareness	significant, positive	-	-	-	significant, positive
Moral norms	-	significant, positive	significant, positive	significant, positive	significant, positive
Personal habits	significant, positive	-	-	-	-
Subjective knowledge	-	-	-	significant, negative	-
Gender ^a^	-	-	-	significant, positive	-
Education ^b^	-	-	significant, negative	-	-
Age (18–25) ^c^	-	significant, positive	significant, positive	-	significant, positive
Age (26–30) ^c^	-	-	significant, positive	-	-
Age (31–40) ^c^	-	-	significant, positive	-	-
Age (51–60) ^c^	-	significant, positive	significant, positive	-	-
Monthly household income (5001–10,000) ^d^	-	significant, negative	-	-	significant, negative
Monthly household income (10,001–20,000) ^d^	-	significant, negative	-	-	significant, negative
Monthly household income (>20,000) ^d^	-	significant, negative	-	-	significant, negative

^a^ = Significance of regression compared to women; ^b^ = Significance of the regression compared to the less educated; ^c^ =Significance of the regression compared to those younger than 18; ^d^ = Significance of the regression compared to those with monthly household income less than 2000 per month. - =no significant effect.

**Table 6 ijerph-20-02185-t006:** Mean willingness to pay according to the DBDC model.

	Coef.	Std. Err.	z	*p* > |z|	[95% Conf. Interval]	
WTP (green food)	81.767	2.491	32.830	0.000	76.885	86.650
WTP (green clothing)	52.502	2.229	23.550	0.000	48.133	56.871
WTP (green travel)	38.908	2.359	16.490	0.000	34.284	43.532
WTP (green housing)	53.150	2.485	21.390	0.000	48.280	58.020
WTP (waste recycling)	37.182	2.239	16.610	0.000	32.794	41.570

## Data Availability

Data is available on request from the authors.

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
