# Peer review of "Public Willingness to Pay for Green Lifestyle in China: A Contingent Valuation Method Based on Integrated Model"

_ijerph, 2023, doi:10.3390/ijerph20032185_

Round 1
Reviewer 1 Report
The topic per se is very interesting and the authors have done a reasonable job of conducting the empirical work. However, it has potential to become more interesting.
My major concerns are listed below.
1. The authors need to present in a more clear way the contribution of this research, given the large body of existing literature on this topic. My question: what is novel?
2. The manuscript inscribed nicely using good communication skill and easy to understand by the audience.
3. The number of sources as cited and their relevance to the study are fine based on the scope and title of the manuscript.
4. The author(s) need to work more on the section methodology and make it understandable. Which are the advantages of employed methodology compared to other techniques?
5. The empirical analysis although interesting, lack the interpretation and economic meaning. I would strongly suggest that the authors add more intuitive explanation.
6. The conclusion section is the one section that needs the little revision. How can these findings really advance future studies?
Author Response
Point 1: The authors need to present in a more clear way the contribution of this research, given the large body of existing literature on this topic. My question: what is novel?
Response 1: Thank you for your valuable comments. We have carefully reviewed the article and referred to a large amount of existing literature on the topic of green lifestyles, and summarized and revised the contribution of the article in the introduction section (in the revision marked in red, lines 104-119). The novelty of this study lies in the fact that it breaks through the shortcomings of previous studies that only focus on single area (for example, organic milk), and starts from all aspects of residents' daily life, divides green lifestyles into five areas: green food, clothing, travel, housing, and waste recycling, and comprehensively examines the drivers that influence Chinese residents to implement green lifestyles and analyzes the differences in residents' real willingness to pay for different green lifestyles.
Point 2: The manuscript inscribed nicely using good communication skill and easy to understand by the audience.
Response 2: Thank you for your approval of this article.
Point 3: The number of sources as cited and their relevance to the study are fine based on the scope and title of the manuscript.
Response 3: Thank you for taking the time to read our articles carefully and for your valuable comments.
Point 4: The author(s) need to work more on the section methodology and make it understandable. Which are the advantages of employed methodology compared to other techniques?
Response 4: Thank you very much for your suggestion, we have reorganized the contents of the methodology section, and in order to visualize the advantages of the DBDC survey method, we have detailed this survey method (in the revision marked in red, lines 277-298) at the beginning of the methodology section.
Point 5: The empirical analysis although interesting, lack the interpretation and economic meaning. I would strongly suggest that the authors add more intuitive explanation.
Response 5: Thank you for raising this valuable question. We have separated the results and discussion sections, with the results section performing the empirical analysis and the discussion section adding more intuitive explanatory language. Besides, the empirical implications of this paper are described in Section 6.2, which may make it easier to understand the analysis of the article.
Point 6: The conclusion section is the one section that needs the little revision. How can these findings really advance future studies?
Response 6: Thank you for pointing this out. We have revisited the concluding section of the paper and separated it into a separate section (section6.3), where we describe in detail the limitations of the paper and the future directions of the research.
Reviewer 2 Report
The paper reports an interesting topic and research that is of high relevance to readers of the journal, however, the following are of much concern:
1) You should explain more in the paper about green life style. You expand literature review about the item.
2) The contribution to theory is high not clear, and the implications of the study should be made clearer.
3) The theoretical framework is unclear. The author should reorganize it. Autor should move the "Contingent Valuation Method" section to the methodology section. You can separate the review of the literature from the concept of the theoretical model.
4) What literature was used to determine the items for the model? What value do the Cronbach's index reach, (the article lacks these values). In each case, it was measured with only one item (sometimes two items). I have doubts to properly examine and recognize factors: e.g. PCB. Personal habits are measured only in energy conservation and emission reduction in daily life. In my opinion the constructs is unsatisfactory.
5) The study has several limitations, which should be more clearly indicated.
6) The research was conducted in China. Therefore, these results cannot be compared to other countries, e.g. European countries.
7) The author should indicate future research directions.
Author Response
Point 1: You should explain more in the paper about green life style. You expand literature review about the item.
Response 1: Thank you for pointing this out. We have explained the green lifestyle in the introduction section (lines 40-44), and considering the understanding of the article. Besides, we have read some more literature on green lifestyle and added a new section (Section2.3) in the literature review section to describe the green lifestyle more visually and in more detail.
Point 2: The contribution to theory is high not clear, and the implications of the study should be made clearer.
Response 2: Thanks to your suggestion, we have added descriptions in the introduction section of the article (in the revision marked in red, lines 104-119) to make the theoretical contributions clearer and described the implications of the article in detail in the conclusion section of the article (section6.1 & section6.2).
Point 3: The theoretical framework is unclear. The author should reorganize it. Autor should move the "Contingent Valuation Method" section to the methodology section. You can separate the review of the literature from the concept of the theoretical model.
Response 3: Thanks to your valuable comments, we have read the theoretical framework section and the methodology section carefully and moved the "Contingent Valuation Method" section to the methodology section (in the revision marked in red, lines277-298) according to your proposal. Besides, we have added a specific explanation of green lifestyle (Section2.2) to the framework section to make the theoretical framework section clearer.
Point 4: What literature was used to determine the items for the model? What value do the Cronbach's index reach, (the article lacks these values). In each case, it was measured with only one item (sometimes two items). I have doubts to properly examine and recognize factors: e.g. PCB. Personal habits are measured only in energy conservation and emission reduction in daily life. In my opinion the constructs is unsatisfactory.
Response 4: 1)As you suggested, we have added a column of references in Table 2 in order to get a more intuitive picture of the supporting literature for each variable and the content of the corresponding questions. Besides, the Cronbach coefficient was calculated to be 0.9, which is also described in the article (section 4.2), indicating that the reliability of the scale is very good, it can properly examine and recognize factors.
2)According to previous studies, subjective norms are divided into descriptive and injunctive norms, attitudes are divided into instrumental and experiential attitudes, and perceptual behavioral control is divided into capacity and autonomy, therefore, these three main variables need two different question items to describe them more comprehensively, and the other four variables can be clearly described by one question item. Therefore, these question items were proposed based on previous studies.
3)Further, we have considered this issue before for the measurement of personal habits, because we have studied a wide range of green lifestyles that cannot be specifically targeted to a particular area, but they all belong to energy saving and emission reduction behaviors, so we use the energy saving and emission reduction questions to measure them.
Point 5: The study has several limitations, which should be more clearly indicated.
Response 5: Thank you very much for your suggestion, we have divided a separate subsection (Section 6.3) in the conclusion section for explaining the limitations of this study (in the revision marked in red, lines 736-752).
Point 6: The research was conducted in China. Therefore, these results cannot be compared to other countries, e.g. European countries.
Response 6: Thank you for raising this valuable question, and we have made a serious consideration. We have separated the results from the discussion section into two sections, and in the discussion section we have compared the differences in willingness to pay for different green living lifestyles in other countries (in the revision marked in red, lines 623-641).
Point 7: The author should indicate future research directions.
Response 7: Thank you very much for your suggestion, we have divided a separate subsection (Section 6.3) in the conclusion section to explain the limitations and future research directions of this study (in the revision marked in red, lines 753-765).
Reviewer 3 Report
The paper addresses an important gap in the literature by focusing on people’s green lifestyle and how much extra money they are willing to pay for it. I think the paper has potential if the authors can improve the overall quality in terms of writing. I have the following comments:
1. To give the reader a brief overview of the study, an “introduction” section should provide a solution to the research problem and a summary of the findings. The authors should use one paragraph to introduce these elements.
2. I found some areas of the manuscript where language use was inefficient. Therefore, it is recommended that authors need to find a native English speaker to proofread the manuscript before submission.
3. In the conclusion section, the authors need to clarify what the limitations of this paper are.
Author Response
Point 1: To give the reader a brief overview of the study, an “introduction” section should provide a solution to the research problem and a summary of the findings. The authors should use one paragraph to introduce these elements.
Response 1: Thank you for pointing this out. Based on your proposal, we have reorganized the language in the introduce section by asking questions - solving problems - presenting results in a sequence (in the revision marked in red, lines 79-99) in order to make the study more intuitive for the reader.
Point 2: I found some areas of the manuscript where language use was inefficient. Therefore, it is recommended that authors need to find a native English speaker to proofread the manuscript before submission.
Response 2: We are very sorry for the trouble caused to your reading of our article. Before that, our article has been polished by professors of English majors and foreign students from English speaking countries. We have also re-read the article and made some modifications.
Point 3: In the conclusion section, the authors need to clarify what the limitations of this paper are.
Response 3: Thank you very much for your suggestion, we have divided a separate subsection (Section 6.3) in the conclusion section to explain the limitations of this study (in the revision marked in red, lines 996-1002).
Reviewer 4 Report
Dear authors
Reading your work was a great opportunity for me to keep improving my knowledge in this research topic. Thank you!
This paper aims to analyse how people recognize the green lifestyle and how 10 much extra money they are willing to pay for it.
The topic addressed is very interesting and important to improve the current knowledge in this research area and for practice and future research. However, I believe that the manuscript could be improved before it can be considered publishable. Please see my comments below. I hope this feedback is helpful for the improvement of the manuscript. Best wishes with your work.
1) One problem observed in the paper is the fact that the authors do not substantiate their statements based on the literature. For example, in the Introduction section at lines 33-39, lines 46-57 and lines 58-59, there are no references to support these statements. Regarding the statements made by the authors in these last lines, it is particularly important that the studies referred to in the following paragraph are cited:
“…which sometimes may lead to potential overestimation or underestimation. In addition, existing studies on the WTP for carbon neutrality mainly focus on aviation and tourism, there is little literature that comprehensively examines the potential carbon-neutral WTP in individual lifestyles”.
In my opinion, this would further reinforce the gap found in the literature.
2) In the Theoretical framework section, at lines 106-116 the citations that support the text are missing. Again, I suggest that the authors substantiate the statements made.
3) The authors state in the Abstract that they base their study on the theory of planned behaviour (TPB). However, TPB suggests that individual attitudes, subjective norms, and perceived behaviour control influence intention; and that intention influences behaviour. In the present study, the authors analyse the influence of these variables on WTP. But, is WTP an environmental behaviour? Or is the variable environmental behaviour "adopting a green lifestyle"? I suggest that this be clarified. Furthermore, there are six variables in the model (excluding the control variables). Three belong to other theoretical models. So, isn't the proposed model a combination of the various models referred to in subsection 2.3? Please clarify this aspect.
Still, regarding the theoretical contributions to the proposed model, it would be interesting to present a table that summarizes the different studies (e.g., Lind; Jakovcevic and Steg) that addressed all the variables presented in the model.
4) For a more fluid reading of the text, I recommend presenting the figures and tables as soon as they are presented. For example, Table 3 should appear right after line 302; Figure 2 following line 307; Figure 3 after line 314; Table 4 after line 489. In addition, the graphs shown in figure 4, if they were presented in the same format (e.g., bar graph) would make it easier to read and compare the results of the different variables analysed.
5) It was unclear whether the scales shown in Table 2 are based on previous studies or created by the authors. I suggest that this information be inserted in the text.
6) Regarding the structure of the manuscript, I believe that the manuscript would benefit from the division of section 4 into two sections: one for presenting the results, and the other for its discussion. This would allow bringing some text present in the conclusions to the discussion. The conclusion in the mode presented in this version is too long and, in my opinion, should be more direct and synthetic.
7) Finally, I suggest a careful reading of the document from a formal point of view. For example, lines 723-729 are out of place.
Best regards
Author Response
Point 1: One problem observed in the paper is the fact that the authors do not substantiate their statements based on the literature. For example, in the Introduction section at lines 33-39, lines 46-57 and lines 58-59, there are no references to support these statements. Regarding the statements made by the authors in these last lines, it is particularly important that the studies referred to in the following paragraph are cited:
“…which sometimes may lead to potential overestimation or underestimation. In addition, existing studies on the WTP for carbon neutrality mainly focus on aviation and tourism, there is little literature that comprehensively examines the potential carbon-neutral WTP in individual lifestyles”.
In my opinion, this would further reinforce the gap found in the literature.
Response 1: As suggested by the reviewer,we have added more references into the introduction part in the revised manuscript. Specifically, we have added papers 7, 8 to lines33-39(in the revision marked in red, lines 33-35 ), papers 9,13,14 to lines46-47(in the revision marked in red, lines 36-57). Moreover, the contents of lines 58-59 were modified, and references 16-20 were added(in the revision marked in red, lines 62-64).
Point 2: In the Theoretical framework section, at lines 106-116 the citations that support the text are missing. Again, I suggest that the authors substantiate the statements made.
Response 2: Thank you for pointing this out. We have added literature 24-28 in lines 106-116(in the revision marked in red, lines 136-148) to confirm the description we have made.
Point 3: The authors state in the Abstract that they base their study on the theory of planned behaviour (TPB). However, TPB suggests that individual attitudes, subjective norms, and perceived behaviour control influence intention; and that intention influences behaviour. In the present study, the authors analyse the influence of these variables on WTP. But, is WTP an environmental behaviour? Or is the variable environmental behaviour "adopting a green lifestyle"? I suggest that this be clarified. Furthermore, there are six variables in the model (excluding the control variables). Three belong to other theoretical models. So, isn't the proposed model a combination of the various models referred to in subsection 2.3? Please clarify this aspect.
Still, regarding the theoretical contributions to the proposed model, it would be interesting to present a table that summarizes the different studies (e.g., Lind; Jakovcevic and Steg) that addressed all the variables presented in the model.
Response 3: Thank you for pointing this out.The environmental behavior you mentioned means implementing a green lifestyle, and the willingness to pay means intention, in the context of green consumer behaviour research, green consumer lifestyle can be predicted by their green purchase intention. Furthermore, As you would expect , the proposed model is a combination of the various models referred to in subsection 2.3. Still, as you suggested, we have added a column for references in Table 2 in order to visualize the supporting literature for each variable and the content of the corresponding questions.
Point 4: For a more fluid reading of the text, I recommend presenting the figures and tables as soon as they are presented. For example, Table 3 should appear right after line 302; Figure 2 following line 307; Figure 3 after line 314; Table 4 after line 489. In addition, the graphs shown in figure 4, if they were presented in the same format (e.g., bar graph) would make it easier to read and compare the results of the different variables analysed.
Response 4: As per your suggestion, we have presented the figures and tables as soon as they are presented except Table 4, because Table 4 is a horizontal page. And we have tried to adjust it which would affect the overall aesthetics of the article layout, and we are sorry we have not adjusted this. For Table 4, we also appreciate your suggestion, the original version of this figure we used a uniform bar chart, but we have taken into account the versatility and aesthetics of the chart, so we have adopted the current version, which we think will increase the overall readability of the paper.
Point 5: It was unclear whether the scales shown in Table 2 are based on previous studies or created by the authors. I suggest that this information be inserted in the text.
Response 5: According to your proposal, we have added a column of references in Table 2 in order to have a more intuitive understanding of the supporting literature for each variable and the content of its corresponding question item.
Point 6: Regarding the structure of the manuscript, I believe that the manuscript would benefit from the division of section 4 into two sections: one for presenting the results, and the other for its discussion. This would allow bringing some text present in the conclusions to the discussion. The conclusion in the mode presented in this version is too long and, in my opinion, should be more direct and synthetic.
Response 6: After discussion, we strongly agree with your proposal. We have divided Section 4 into Results (Section 5) and Discussion (Section 6) and placed a part of the description from the conclusion in the discussion, thus avoiding that the conclusion is too long to affect the reader's understanding of the article.
Point 7: Finally, I suggest a careful reading of the document from a formal point of view. For example, lines 723-729 are out of place.
Response 7: We were really sorry for our careless mistake. Thank you for your reminder. we have revisited the article and removed line 723-729.
Round 2
Reviewer 4 Report
Good work on improving the manuscript!